# A Smart Contract-Based P2P Energy Trading System with Dynamic Pricing on Ethereum Blockchain

**DOI:** 10.3390/s21061985

**Published:** 2021-03-11

**Authors:** Jae Geun Song, Eung seon Kang, Hyeon Woo Shin, Ju Wook Jang

**Affiliations:** Department of Electronic Engineering, Sogang University, Seoul 04107, Korea; skj1080@sogang.ac.kr (J.G.S.); kes617@sogang.ac.kr (E.s.K.); sinhyunwoo@sogang.ac.kr (H.W.S.)

**Keywords:** smart contract, peer-to-peer energy trading, blockchain, Ethereum, dynamic pricing, microgrids

## Abstract

We implement a peer-to-peer (P2P) energy trading system between prosumers and consumers using a smart contract on Ethereum blockchain. The smart contract resides on a blockchain shared by participants and hence guarantees exact execution of trade and keeps immutable transaction records. It removes high cost and overheads needed against hacking or tampering in traditional server-based P2P energy trade systems. The salient features of our implementation include: 1. Dynamic pricing for automatic balancing of total supply and total demand within a microgrid, 2. prevention of double sale, 3. automatic and autonomous operation, 4. experiment on a testbed (Node.js and web3.js API to access Ethereum Virtual Machine on Raspberry Pis with MATLAB interface), and 5. simulation via personas (virtual consumers and prosumers generated from benchmark). Detailed description of our implementation is provided along with state diagrams and core procedures.

## 1. Introduction

With the introduction of renewable energy resources, traditional energy consumers are becoming “prosumers” who use photovoltaic panels or wind power generators to generate energy and make profit by selling the surplus energy after consumption to neighboring consumers. The direct energy trade among prosumers and consumers is called peer-to-peer (P2P) energy trading. A node participating in a P2P energy trading system stores generated energy in an energy storage system (ESS) and the smart meter may record change of energy due to generation, consumption, out-flow and in-flow.

However, without intervention by the trusted third party, it is impossible or hard to guarantee trust between participants, determine the price of energy trading, or fulfill the agreement automatically or forcibly in conventional P2P energy trading systems [1]. In addition, these server-based systems are vulnerable to hacking and tampering unless costly firewalls are installed. The cost and maintenance overheads resulting from this security enforcement may be formidably high for small-scale P2P trading within a microgrid. We leverage blockchain technology [2] to remove the cost and the overhead while guaranteeing integrity of trading records. The salient features of our P2P energy trading system are as follows:
(1)Dynamic pricing for automatic balancing of total supply and total demand within a microgrid: We assume that our energy trading system within a microgrid will help prosumers and consumers to trade small amounts of energy in each short trading period, for example, an hour. Under this assumption, it would be cumbersome for prosumers and consumers to bid or ask in every trade for each period. To avoid this hassle, in our system, a single price is determined for each period as a function of total demand and total supply submitted. The price increases/decreases per each period depend on the ratio of total demand over total supply. The increased/decreased price will encourage/discourage supply/demand for prosumers (consumers) in the next trading period. The speed of convergence is adjustable for faster convergence or stable operation through manipulation of the pricing formula. This mechanism will help our trading system automatically achieve equilibrium (total supply = total demand) or operate close to it without any intervention by a third party.(2)Prevention of double sale: Since energy is traded online, it is imperative to guarantee that the same energy is not sold more than once. For this, we introduce an energy ownership structure and implement it inside a smart contract [3] on Ethereum blockchain [4]. The energy ownership structure guarantees integrity for every state change of energy: “Injected”, “On-board for sale”, “Matched” and so on. Change of states is only allowed for qualified participants and protected from any hacking or tampering by any unauthorized party. For example, injection of energy is only verified by DSO (distribution system operator) and hence only the DSO is qualified to update the corresponding state with its private key. Traditionally, the DSO is responsible for almost everything for trading on the energy market and is vulnerable to hacking or tampering. In our system, we replace the DSO by Ethereum blockchain as much as possible, excluding indispensable parts such as transmission, confirmation of injection and pricing policy. Matching, payment, prevention of double sale, et cetera, are automatically and forcefully executed by a smart contract, free from hacking or tampering.(3)Automatic and autonomous operation: The trading procedure is implemented as a smart contract on Ethereum and hence trade in each period is performed automatically and autonomously. At the start of each trading period, prosumers and consumers send request_to_sell and request_to_buy, respectively, to the smart contract. The consumers deposit enough money to cover their purchase to the smart contract. The smart contract collects all the requests and computes an energy price according to a preset formula. Matching and clearing are also performed by the smart contract. Thus, all the procedures are automatic and autonomous, requiring neither intervention of third parties nor costly firewalls. The smart contract performs as an escrow between prosumer, consumer and DSO to ensure that the promised transaction is actually delivered. In our current implementation, the DSO is responsible for transmission and withdrawal of power. The DSO uses its private key to create a digital signature which guarantees transmission and withdrawal of power. As for payment, the smart contract performs as an escrow to verify the delivery of power by the DSO and to ensure that the payment is done accordingly.(4)Experiment on a testbed: Many known works in the literature on trading on blockchain sketch their schemes without actual implementation. We perform an experiment on a testbed using Raspberry Pis as prosumers, consumers and a DSO. All nodes have their own virtual machine called Ethereum Virtual Machine (EVM) [5] on which a smart contract is executed. Prosumers, consumers and DSO use Node.js and web3.js API to control Geth (a command line interface to access EVM). The GUI is built with MATLAB [6].(5)Simulation via persona: We borrow from existing energy production and consumption data to create virtual prosumers and consumers to experiment on our testbed. We generate personas in such a way that they respond to price increase/decrease to increase/decrease supply or decrease/increase demand, correspondingly. Their behaviors are programmable to suit any scenario.

The rest of the paper is organized as follows: In Section 2, we summarize related works. Our P2P energy trading system with dynamic pricing mechanism and state diagram is presented in Section 3. Implementation on a private Ethereum blockchain is described in Section 4. The experiment using a testbed with persona is shown in Section 5 and Section 6 concludes this paper.

## 2. Related Works

### 2.1. Pricing Models

The double auction is a mechanism involving both buyers and sellers, which simultaneously participate in the bidding process and are allocated individual shares of the resource [7]. An aggregator communicates with the other agents about the supply, demand and bidding price that each buyer is willing to pay and implements a distributed double auction algorithm to determine price. Asynchronous double auction mechanism, another double auction model proposed in [8], can be used as a P2P energy trading technique. The matching priority and the prices are determined by the amount of energy and bidding price possible between a preset upper limit and a lower limit. In double auction models, all participants manually bid or ask, which may be cumbersome or time-consuming.

Several dynamic pricing models are proposed for smart grids: quadratic cost function (QCF) [9], usage-based dynamic pricing (UDP) [10], distributed demand response (D2R) [11] and distributed dynamic pricing (D2P) [12]. QCF [9] uses a neural network for piecewise QCF. If this model is applied to pricing, the microgrid decides the price only depending on the supply. As a result, consumers may have to pay higher prices even though the total demand within the microgrid is low. In UDP [10], real-time pricing is performed as a quadratic function with energy demand as a variable during peak hours. Otherwise, the price is fixed. In [11], the price is the *k*-square of energy demand. The cost incurred by consumers depends directly on the energy demand even though the microgrids have excess surplus energy to serve. As a result, consumers may not consume more energy, even at lower prices, thus wasting excess surplus energy.

To consider both supply and demand, in [12], real-time pricing is performed from the minimum price and the difference between total supply and total demand. The price decreases with an increase in the supply from the prosumers while the demand from the consumers is either fixed or decreased. On the other hand, the price increases with an increase in the demand, while the supply is either fixed or decreased. Thus, prosumers and consumers can control supply or demand according to the price. However, the price is determined only depending on the difference. So, the price may be the same whether the ratio of demand over supply changes or not, if the difference remains same.

Going on further from the proposed D2P model, Chekired, Khoukhi and Mouftah [13] propose a pricing model that considers both the difference and the ratio between the total demand and total supply in round t. The dynamic real-time price changes according to the variation in the difference and the ratio by using exponential and arctangent functions. However, this pricing model is not very adaptive to the change in demand/supply ratio as we show in Section 3. We devise a dynamic pricing scheme which is more adaptive and adjustable in controlling the speed of convergence to equilibrium (total demand = total supply). Details will be provided in Section 3.

### 2.2. Blockchain Technology

Blockchain [14] is a distributed ledger over a P2P network where the encrypted data is shared and recorded to all participating nodes in a chronological order. The transactions are collected into a block when they are considered valid through a validation process known as a proof of work (POW). Ethereum [4] is a programmable blockchain for building decentralized applications, allowing anyone to write smart contracts [3]. In Ethereum we can create our own arbitrary rules for ownership, transaction formats and station transition functions. In this paper, we implement a smart contract-based P2P energy trading system on Ethereum blockchain.

The smart contract is a blockchain-based program that encodes the conditions for fulfillment of an agreement between participants. It automatically executes the agreement when the conditions are met. It can be written in Solidity language [15] and can be built on top of the Ethereum platform. It should be more like an autonomous agent that resides in the Ethereum execution environment. Thus, it always executes a specific code when a message or transaction is transmitted and has direct control over its own balance and its own key/value store to keep track of persistent variables [4].

The term transaction is used in Ethereum to refer to a signed data package that stores records on the blockchain and is sent from an externally owned account. If the smart contract is mined to a block, residing on the blockchain, it has a unique address (the smart contract address). It is of the same type as the externally owned account and controlled by the smart contract code. All nodes can execute the smart contract function by sending a transaction referring the smart contract address. When the smart contract address receives a transaction, its code activates, allowing it to read from or write to internal storage and perform various actions.

In addition, the smart contract has an object called event. The event is an abstraction of the Ethereum logging/event-watching protocol. Log entries provide the smart contract address, a series of up to four topics and some arbitrary length binary data [15]. If an event is called by the smart contract, all nodes can detect and watch the event because they run and share the same state of the smart contract. Thus, participating nodes execute the function and action in accordance with the results of the smart contract event.

### 2.3. Blockchains in Energy Trading

R. Skowronski [16] proposed open-trade through blockchain and a hierarchy-based control of flows for the first time. R. Skowronski [17] tackles the problematics of aiding cyber-physical systems through blockchain-based VMs (Virtual Machines).

Priwatt [18] is a decentralized P2P energy trading system, which provides anonymous communication channels and the means to form agreements without trusting other parties by using Bitmessage [19] and multi-signature techniques. It is built upon Bitcoin [20] blockchain. The system can be applied to the microgrids. The nodes participating in the network are assumed to be prosumer, consumer and distribution system operator (DSO), which confirms injection of energy and is responsible for actual transmission of energy ensuing trade.

When a prosumer injects the surplus energy to sell, the DSO sends a private message to the prosumer with two secret keys which verify the prosumer’s ownership and can be used as a lock to prevent double spending. Participants use auction panel and send a private message to negotiate energy trading. When matching between a prosumer and a consumer is complete, the energy agreed for the sale is locked and the prosumer creates a 2-of-3 multi-signature transaction that requires two out of three signatures (i.e., prosumer, consumer and DSO) to be executed.

Then the consumer specifies the input tokens and signs the transaction. After receiving the payment, the prosumer sends the energy ownership to the consumer [18]. In the Priwatt system, if there is a dispute between participants, DSO mediates the issue to resolve. In addition, trading procedures such as payment, change of energy ownership and execution of trading contracts are not implemented automatically because the system is based on the Bitcoin system.

In our system, based on Ethereum, we devise a dynamic pricing algorithm which balances between demand and supply within a microgrid. We also design a state diagram for our energy trading procedure. We implement our dynamic pricing algorithm and trading procedures in a smart contract on Ethereum. The smart contract in our Ethereum implementation removes any disputes and executes energy trading procedures automatically. It also prevents double sale problems by keeping the changes in energy ownership as trading is performed inside a structure array embedded in the smart contract, and hence is free from any tampering.

Personas are archetypal users who embody the goals and aspirations of real users in an easy-to-assimilate and personable form [21]. They have attributes that represent a specific person or group and act like them. Recently, personas have been widely used for designing user experiences.

In our study, we set up virtual personas that act according to given conditions related to energy trading. They act as prosumers or consumers and make virtual transactions in the system. The prosumers’ supply depends on the prices. Furthermore, demands of consumers varies depending on the price. Using these virtual personas, simulation of the pricing model proposed in this study is performed.

## 3. The Proposed P2P Energy Trading System with Dynamic Pricing

### 3.1. Dynamic Pricing

The main objective of our dynamic pricing algorithm is to balance supply and demand among prosumers and consumers within a microgrid. For example, if total demand in a trading round exceeds the total supply, then the price of each energy unit will be increased to discourage demand. Our dynamic pricing enables our energy trading system to reach equilibrium in which total supply matches total demand within a microgrid. For microgrids of small or medium size, we find bidding algorithms [7,8] unpractical since they require human monitoring and involvement per each trading period (e.g., 1 h or 30 min) for small amounts of energy. Instead, we choose to use one price per each trading round, which is determined only on the total supply and total demand submitted at the start of each trading period.

Let us denote total supply and total demand at the start of trading round t as $ES\left(t\right)$ and $ED\left(t\right)$ as shown in Equations (1) and (2):(1)$$ES\left(t\right)=\overset{n_{p}}{\underset{i=1}{\sum }}S_{i}\left(t\right)\;\left(S_{i}\left(t\right)\geq 0\right)$$
(2)$$ED\left(t\right)=\overset{n_{c}}{\underset{j=1}{\sum }}D_{j}\left(t\right)\;\left(D_{j}\left(t\right)\geq 0\right)$$
where $S_{i}\left(t\right)$ is the supply of prosumer *i*; $D_{j}\left(t\right)$ is the demand of consumer *j*; and $n_{p}$ and $n_{c}$ are the numbers of prosumers and consumers.

$R\left(t\right)$ and $D\left(t\right)$ denote the ratio and the difference between total demand and total supply, respectively, as in Equations (3) and (4).
(3)$$R\left(t\right)=\frac{ED\left(t\right)}{ES\left(t\right)}$$
(4)$$D\left(t\right)=ED\left(t\right)-ES\left(t\right)$$

Chekired, Khoukhi and Mouftah [13] have proposed a dynamic real-time pricing using $R\left(t\right)$ and $D\left(t\right)$ as in Equation (5).
(5)$$p\left(t\right)=\left\{tan^{-1}\left(e^{D\left(t\right)}\right)+{\left(tan^{-1}R\left(t\right)\right)}^{10}\right\}+p_{min}$$

We observe from (5) that if total supply ($ES\left(t\right)$) greatly exceeds total demand ($ED\left(t\right)$), the price is reduced to $p_{min}$, which is the minimum price (usually imposed by the DSO or the management of the microgrid to cover basic expenses for generating energy). The price $p\left(t\right)$. varies between ($p_{min},\left(\frac{\pi }{2}\right)+{\left(\frac{\pi }{2}\right)}^{10}+\;p_{min}$). To conform to pricing guides by electrical companies, it uses exponential and arctangent functions, keeping the resulting price in a given interval [13]. The price is increased rather slowly until demand greatly exceeds supply. The price curve is not suitable for our purpose of balancing the demand and supply since the curve is not symmetric. Later we will compare our dynamic pricing algorithm against this pricing. 

### 3.2. The Proposed Dynamic Pricing Algorithm

Our dynamic pricing algorithm is represented in Equation (6) and Figure 1.
(6)$$p\left(t\right)=\frac{2}{\pi }\left(p_{con}\right)\cdot tan^{-1}({\left(\ln R\left(t\right)\right)}^{k})\;+\;p_{balance}$$

$p_{balance}$ is a price when total demand is equal to total supply ($R\left(t\right)$=1). We use $p_{con}$ to determine the range of price ($p_{balance}-p_{con}$, $p_{balance}+p_{con}$).

Since $\underset{R\left(t\right)\to 0}{\lim}tan^{-1}\left({\left(\ln R\left(t\right)\right)}^{k}\right)=-\frac{\pi }{2}$ and $\underset{R\left(t\right)\to \infty }{\lim}tan^{-1}\left({\left(\ln R\left(t\right)\right)}^{k}\right)=\frac{\pi }{2}$, the price $p\left(t\right)$ ranges from $p_{balance}-p_{con}$ to $p_{balance}+p_{con}$. If total demand matches total supply, in other words, $R\left(t\right)$ is 1, $p\left(t\right)$ is set to $p_{balance}$. In this way, we may choose the balance price as well as the minimum price and the maximum price. We think that one price per trading round reduces the burden of prosumers and consumers. They may choose their own way in reacting to this price change. They may increase supply or reduce demand according to their preset rules. The rules can be represented as a line, curve or step function of supply/demand against price. For DSO, the exponent “*k*” in Equation (6) can be used to control the price curve as illustrated in Figure 2 for *k* = 3, 5 and 7. In Section 5 we show how this enables DSO to make a tradeoff between convergence speed and exactness of balancing to suit its purpose. 

Figure 1 compares our pricing model in Equation (6) against Chekired et al. [13] in Equation (5). We assume that $tan^{-1}\left(e^{D\left(t\right)}\right)$ of Equation (5) is 0 because it tends to be a very small number. In our model, When $R\left(t\right)$ is 1, the price $p\left(t\right)$ becomes 100 which is preset as $p_{balance}$. Additionally, the price exhibits saturation near the maximum or minimum price. On the other hand, in Chekired et al. [13] as Equation (5) continues to increase price even if $R\left(t\right)$ is more than $10^{2}$ as in Figure 1b. Furthermore, when $R\left(t\right)$ is smaller than $10^{0}$, price hardly changes even when demand further increases. It shows asymmetry and it is hard to set up saturation near maximum or minimum. Our pricing model exhibits symmetry and saturation near maximum and minimum.

Figure 2 shows $p\left(t\right)$ against $R\left(t\right)$ with *k* = 3, 5 or 7, respectively. If $R\left(t\right)$ is very small, $p\left(t\right)$ is close to the minimum price (=70) and if $R\left(t\right)$ is large, $p\left(t\right)$ converges to the maximum price (=130). When $R\left(t\right)$ is 1, $p\left(t\right)$ is 100, which is the $p_{balance}$. Price changes symmetrically against $R\left(t\right)$ in log scale. In addition, the slope of the price curve changes depending on the *k*. Our proposed pricing model can easily choose $p_{balance}\;\mathrm{and}$
$p_{con}$ as needed by the DSO, utility companies, government authorities or microgrid managements.

### 3.3. The State Diagram Representation and Solidity Program

For state diagram representation of our trading system, we assume that the participating nodes in the energy trading blockchain within a microgrid are prosumers, consumers and the DSO. Prosumers increase/decrease energy supply in reaction to price increase/decrease. A consumer buys energy from prosumers for their own need. Consumers reduce/increase energy consumption in reaction to price increase/decrease. The smart contract uses Equations (1)–(3) and (6) to determine energy price for each trading period (e.g., hour, day). The DSO acts as an operator or manager of the blockchain network. It is for energy transmission and a smart contract for energy trading. It creates, upgrades and distributes a smart contract. Furthermore, DSO adjusts the exponent “*k*” in Equation (6) to determine how quickly it will reach equilibrium.

We define states for each prosumer or consumer and show how the states change in each phase. Refer to Algorithm 1 with Table 1 at the end of Section 4 which shows our implementation of trading procedure in a Solidity-like pseudo-code. Solidity [15] is a widely used programming language for developing smart contracts on Ethereum. The states are implemented using enum type in Solidity. The enum allows programmers to define a set of allowed members [15]. We choose this feature to define five states of energy for each prosumer or consumer as “register”, “injected”, “board”, “match” and “purchased” (line 8 of Algorithm 1). Figure 3 and Figure 4 illustrate state changes for prosumers and consumers. $P_{i}\left(t\right)\;or\;C_{j}\left(t\right)$ denotes the state of prosumer *i* or that of consumer *j* respectively, within the microgrid in phase *t*.

Any prosumer or consumer intending to trade energy should register with the private Ethereum blockchain associated with the microgrid. It enters phase 0 after registration. The registration is performed by sending a transaction which executes the “Register” function (line 12 of Algorithm 1). The function txAddr $P_{i}$.sendtx(Resister, timestamp) sends the transaction to smcAddr which is the address of the smart contract responsible for energy trading. Line 13 or line 17 initializes state vectors ($O_{pi}$ and $O_{cj}$) representing amounts of energy in the course of energy trading.

$P_{i}\left(t\right)$ or $C_{j}\left(t\right)$ has a vector of three states, respectively. $\left\{I_{i}\left(t\right),\;B_{i}\left(t\right),\;M_{i}\left(t\right)\right\}$ represents injected energy, energy on board for sale and energy matched for trading for a prosumer *i*. $\left\{B_{j}\left(t\right),\;M_{j}\left(t\right),\;P_{j}\left(t\right)\right\}$ represents energy intended to buy, energy matched for trading and energy paid for a consumer *j*.

The vectors of states are defined by using struct EnergyOwnership in lines 9–10. Lines 13 and 17 show instantiation of state vectors for prosumer *i* and consumer *j*. The events invoked by the smart contract on receipt of “Register” transaction from joining prosumer *i* and consumer *j* initialize the vectors of states for prosumer (line 14) and consumer (line 18), respectively.

Prosumer *i* injects the surplus energy $E_{i}$ into the microgrid by executing the INJECTENERGY procedure (lines 21–26 of Algorithm 1 at the end of Section 4). In line 22, prosumer *i* sends an encrypted message to the DSO (msgAddr*D*). The message contains the amount of injected energy and is signed by the private key of the prosumer *i*. The DSO verifies the signature, checks if the claimed injection is complete and sends a confirming transaction (signed by the private key of the DSO) to the smart contract (line 23). The smart contract verifies the signature of the DSO (line 24) to invoke an event which updates the state vector for prosumer *i* with injected energy $E_{i}\;$(line 25).

Note that only the DSO, which physically controls transmission of energy, can confirm the injection or delivery of energy. Thus, in our implementation of the energy trading smart contract, we require the signature of the DSO to update the amount of injected energy in the state vector for prosumer *i*. Therefore, hackers, whether in or out of the blockchain, cannot tamper with it. After injection, the state of the prosumer *i* changes to phase 1, $P_{i}\left(1\right),$ and the amount of injected energy $I_{i}\left(1\right)$ is set to $E_{i}$.

Prosumer *i* publishes its intent to sell energy of $S_{i}\left(t\right)$ on board by executing the “RequestSell” function (line 30). The state changes to $P_{i}\left(2\right)$. $B_{i}\left(2\right)$, on-board energy for sale, is set to $S_{i}\left(t\right)$ and $I_{i}\left(2\right)$ is decreased to $E_{i}\;-\;S_{i}\left(t\right)$. When matching phase begins by the “Matching” function (line 41), the state changes to $P_{i}\left(3\right)$. The amount of matched energy (actually sold) is $S_{m}{}_{i}$ as in lines 42–49. The unmatched energy (remaining after sale), $S_{i}\left(t\right)\;-\;S_{m}{}_{i},$ is returned to “injected”. Thus, the amount of energy with the state vector $I_{i}\left(3\right)$ is $E_{i}\;-\;S_{m}{}_{i}$. When the matching is complete and the DSO executes the “Trade” function (line 57), prosumer *i* receives payment $S_{m}{}_{i}\cdot p\left(t\right)$ from the smart contract (line 59) and changes to $P_{i}\left(4\right)$. Prosumer *i* can inject surplus energy $S_{i}\left(t\right)\;-\;S_{m}{}_{i}$ again, i.e., $I_{i}\left(4\right)$ = $E_{i}\;-\;S_{m}{}_{i}$.

Consumer *j* puts its intent to buy on board to purchase the amount of energy $D_{j}\left(t\right)$ by sending the transaction that executes the “RequestBuy” function (line 35). Consumer *j* deposits the amount of $D_{j}\left(t\right)\cdot p_{max}$ to the smart contract by sending the transaction that executes the “Transfer” function (line 37). The state of consumer *j* changes to $C_{j}\left(1\right)$ with $B_{j}\left(1\right)$ = $D_{j}$. In the matching phase, the “Matching” function is executed (line 41) and the state of consumer *j* changes to $C_{j}\left(2\right)$ with matched amount of energy $M_{i}\left(2\right)$ = $D_{m}{}_{j}$ (lines 42–49 for calculation). 

When the matching is complete and the DSO executes the “Trade” function (line 57), the consumer receives the refund of $D_{j}\left(t\right)\cdot p_{max}$ — $D_{m}{}_{j}\cdot p\left(t\right)$ from the smart contract (line 62) where $p\left(t\right)$ is the price for phase *t* as determined by Equation (6). The state of consumer *j* changes to $C_{j}\left(3\right)$. Consumer *j* obtains the ownership of the amount of energy $P_{j}=D_{m}{}_{j}$. Consumer *j* can receive the energy from the DSO by using this ownership and then the state changes to $C_{j}\left(4\right)$.

**Algorithm 1:** Energy Trading Algorithm
**1.** **procedure** ENERGYTRADING**2.** txAddr← hash(public key), msgAddr
← hash(public key of whisper)**3.** ${\mathrm{Prosumer}}_{i}$←. $\mathrm{txAddr}P_{i}$*,*$\mathrm{msgAddr}P_{i}$, $O_{pi}$, $E_{i}$, $S_{i}\left(t\right)$, $S_{m}{}_{i}$$(1\leq i\leq n_{p}$)**4.** ${\mathrm{Consumer}}_{j}$$\leftarrow \mathrm{txAddr}C_{j}$*,*$\mathrm{msgAddr}C_{j}$*,*$O_{cj}$, $D_{j}\left(t\right)$*,*$D_{m}{}_{j}$$(1\leq j\leq n_{c}$)**5.** DSO ←txAddrD, msgAddrD*,*
$ES\left(t\right)$*,*
$ED\left(t\right)$**6.** Smart Contract*(SmC)*
←
smcAddr**7.** **procedure** REGISTER(${\mathrm{Prosumer}}_{1},\ldots ,{\mathrm{Prosumer}}_{n_{p}},{\;\mathrm{Consumer}}_{1},\ldots ,{\;\mathrm{Consumer}}_{n_{c}})$**8.** 
**enum**
$\mathrm{State}\left\{\mathrm{register},\;\mathrm{injected},\;\mathrm{board},\;\mathrm{match},\;\mathrm{purchased}\right\}$
**9.** **struct**EnergyOwnership{**10.** **address**account;**uint**amount;**State**state;**uint**timestamp;}**11.** 
**if**
$\mathrm{txAddr}\in \mathrm{txAddr}P_{i}$
**then**
**12.** $\mathrm{txAddr}P_{i}.\mathrm{sendtx}\;(Register,\;timestamp$) ⇒ smcAddr**13.** 
$\mathrm{EnergyOwnership}\left[\;\right]$
$O_{pi};$
**14.** *SmC.*event($O_{pi}\leftarrow \left(\mathrm{txAddr}P_{i},0,register,timestamp\right)$)**15.** 
**else if**
$\mathrm{txAddr}\in \mathrm{txAddr}C_{j}$
**then**
**16.** $\mathrm{txAddr}C_{j}.\mathrm{sendtx}\;(Register,\;timestamp$) ⇒ smcAddr**17.** 
$\mathrm{EnergyOwnership}\left[\;\right]$
$O_{cj};$
**18.** *SmC.*event ($O_{cj}\leftarrow \mathrm{txAddr}C_{j},0,register,timestamp)$)**19.** 
**end if**
**20.** 
**end procedure**
**21.** **procedure** INJECTENERGY($E_{i}$)**22.** $\mathrm{msgAddr}P_{i}.\mathrm{msg}$ ($Inject,\;\mathrm{txAddr}P_{i},\;E_{i})$ ⇒ msgAddrD**23.** $\mathrm{txAddr}D.\mathrm{sendtx}\;\left(Inject,\;\mathrm{txAddr}P_{i},\;E_{i},\;timestamp\right)$ ⇒ smcAddr**24.** *SmC*.require $\left(\mathrm{msg}.\mathrm{sender}==\mathrm{txAddr}D\right)$**25.** *SmC*.event($O_{pi}\leftarrow \left(\mathrm{txAddr}P_{i},\;\;E_{i},\;\;injected,\;\;timestamp\right)$)**26.** 
**end procedure**
**27.** **procedure** AGGREGATION **(**${\mathrm{Prosumer}}_{1},\ldots ,{\mathrm{Prosumer}}_{n_{p}},{\;\mathrm{Consumer}}_{1},\ldots ,{\mathrm{Consumer}}_{n_{c}}$ during ∆t)**28.** txAddrD.sendtx(Roundstart, timestamp) ⇒ smcAddr**29.** 
$order\;elements\left(state\;:injected\right)$
$for\;all\;O_{pi}$
**30.** $\mathrm{txAddr}P_{i}.\mathrm{sendtx}$$(RequestSell,\;S_{i},\;timestamp$) ⇒ smcAddr**31.** **if***SmC*.validate $\left(S_{i}\leq E_{i},timestamp\in ∆t\right)=true$**then****32.** *SmC*.event($O_{pi}\leftarrow \left(\mathrm{txAddr}P_{i},\;\;S_{i}\left(t\right),\;\;board,\;\;timestamp\right)$)**33.** *SmC*.event($O_{pi}\leftarrow \left(\mathrm{txAddr}P_{i},\;\;E_{i}=E_{i}-S_{i},\;\;injected,\;\;timestamp\right)$)**34.** 
**end if**
**35.** $\mathrm{txAddr}C_{j}.\mathrm{sendtx}$ ($RequestBuy,D_{j}\left(t\right),\;timestamp$) ⇒ smcAddr**36.** **if***SmC*.validate $\left(D_{j}\left(t\right)\cdot p_{max}\leq \mathrm{txAddr}C_{j}.\;balances,timestamp\in ∆t\right)=true$**then****37.** $\mathrm{txAddr}C_{j}.\mathrm{sendtx}$ ($Transfer,\;\mathrm{smcAddr},\;\;D_{j}\left(t\right)\cdot p_{max},\;timestamp$) ⇒ smcAddr**38.** 
**end if**
**39.** 
**end procedure**
**40.** **Procedure** MATCHING **(**${\mathrm{Prosumer}}_{1},\ldots ,{\mathrm{Prosumer}}_{n_{p}},S_{1}\left(t\right),\ldots ,S_{n_{p}}\left(t\right),{\;\mathrm{Consumer}}_{1},\ldots ,$**41.** ${\mathrm{Consumer}}_{n_{c}},\;D_{1}\left(t\right)$,…,$D_{n_{c}}\left(t\right)$ about ∆t)**42.** txAddrD.sendtx (Matching,  ∆t,  timestamp) ⇒ smcAddr
**43.** 
$ES\left(t\right)\leftarrow \overset{n_{p}}{\underset{1}{\sum }}S_{i}\left(t\right),\;\mathrm{ED}(t)\leftarrow \overset{n_{c}}{\underset{1}{\sum }}D_{j}\left(t\right)$
**44.** 
$q=ED\left(t\right)/ES\left(t\right)$
**45.** 
**i**
**f**
*E*
$S\left(t\right)\geq ED\left(t\right)$
**46.** $S_{m}{}_{i}=q\cdot S_{i}\left(t\right)$, $D_{m}{}_{j}=D_{j}\left(t\right)$, $\overset{n_{p}}{\underset{1}{\sum }}S_{m}{}_{i}=\overset{n_{c}}{\underset{1}{\sum }}D_{m}{}_{j}$**47.** 
**else if**
$ED\left(t\right)>ES\left(t\right)$
**48.** $S_{m}{}_{i}=S_{i}\left(t\right)$, $D_{m}{}_{j}=D_{j}\left(t\right)$/q, $\overset{n_{p}}{\underset{1}{\sum }}S_{m}{}_{i}=\overset{n_{c}}{\underset{1}{\sum }}D_{m}{}_{j}$**49.** 
**end if**
**50.** *SmC*.delete($O_{pi}\leftarrow \left(\mathrm{txAddr}P_{i},\;\;S_{i}\left(t\right),\;\;board,\;\;timestamp\right)$)**51.** *SmC*.event($O_{pi}\leftarrow \left(\mathrm{txAddr}P_{i},\;\;S_{m}{}_{i},\;\;match,\;\;timestamp\right)$)**52.** *SmC*.event($O_{pi}\leftarrow \left(\mathrm{txAddr}P_{i},\;\;S_{i}\left(t\right)-S_{m}{}_{i},\;\;injected,\;\;timestamp\right)$)**53.** *SmC*.event($O_{cj}\leftarrow \left(\mathrm{txAddr}C_{j},\;\;D_{m}{}_{j},\;\;match,\;\;timestamp\right)$)**54.** 
**end procedure**
**55.** **procedure** TRADE ENERGY(${\mathrm{Prosumer}}_{1},\ldots ,{\mathrm{Prosumer}}_{n_{p}},{\;\mathrm{Consumer}}_{1},\ldots ,{\mathrm{Consumer}}_{n_{c}}$)**56.** *Pricing by DSO*: $p=\frac{2}{\pi }\cdot p_{con}\cdot \tan^{-1}{\left(\ln(\frac{ED\left(t\right)}{ES\left(t\right)})\right)}^{k}+p_{balance}$**57.** txAddrD.sendtx (Trade,  p,  ∆t,  timestamp) ⇒ smcAddr**58.** 
**for**
i=1
**to**
$n_{p}$
**59.** 
$\mathrm{txAddr}P_{i}.balances=\mathrm{txAddr}P_{i}.balances+S_{m}{}_{i}\cdot p\left(t\right)$
**60.** smcAddr.sendtx ($Transfer,\;\mathrm{smcAddr},\;\;S_{m}{}_{i}\cdot p\left(t\right),\;timestamp$) ⇒ $\mathrm{txAddr}P_{i}$**61.** *SmC*.delete($O_{pi}\leftarrow \left(\mathrm{txAddr}P_{i},\;\;S_{m}{}_{i},\;\;match,\;\;timestamp\right)$)**62.** 
**for**
j=1
**to**
$n_{c}$
**63.** 
$\mathrm{txAddr}C_{j}.balances=\mathrm{txAddr}C_{j}.balances+\left(D_{j}\left(t\right)\cdot p_{max}-D_{m}{}_{j}\cdot p\left(t\right)\right)$
**64.** smcAddr.sendtx ($Transfer,\;\mathrm{smcAddr},\;\;D_{j}\left(t\right)\cdot p_{max}-D_{m}{}_{j}\cdot p\left(t\right),\;timestamp$) ⇒ $\mathrm{txAddr}C_{j}$**65.** *SmC*.delete($O_{cj}\leftarrow \left(\mathrm{txAddr}C_{j},\;\;D_{m}{}_{j},\;\;match,\;\;timestamp\right)$)**66.** *SmC*.event($O_{cj}\leftarrow \left(\mathrm{txAddr}C_{j},D_{m}{}_{j},purchased,timestamp\right)$)**67.** 
smcAddr.balances=0
**68.** 
**end procedure**
**69.** 
**end procedure**



## 4. The Proposed P2P Energy Trading Implementation on a Private Ethereum Blockchain

Figure 5 shows the software architecture of our P2P energy trading system. All nodes (prosumers, consumers and the DSO) have their own virtual machine called Ethereum Virtual Machine (EVM) [5] on which a smart contract is executed. Each node uses Node.js and web3.js API to conveniently control Geth. Geth [22] is a command line interface for running a full Ethereum node implemented in Go language and is able to access the EVM. To execute a smart contract on EVM, a node needs a Solidity compiler called solc.js API to compile the smart contract. 

Smart contracts on Ethereum can be written in Solidity [15] language. Participating nodes send transactions including which functions to execute, parameters required to execute the function, compiled bytecode of the smart contract and the corresponding smart contract address. When the transactions are mined into a block, all nodes execute the smart contract functions with the given parameters. Then the EVMs of all nodes run the function and maintain the same state.

Ethereum blockchain provides an identity-based messaging system which is called Whisper [23]. Whisper provides anonymity and privacy in a trustless network via broadcasting encrypted messages in messaging streams. Every node may have symmetric and asymmetric keys and use an envelope which is a packet sent and received in Whisper. All messages are encrypted either symmetrically or asymmetrically and nodes use the keys to decrypt received envelopes [23].

Note that messages are transmitted through Whisper without causing traffic on the Ethereum block since they are not mined into a block. In this paper, Whisper messaging is used to generate an element with the state of “Injected” when a prosumer injects the surplus energy. Whisper can be activated on Geth, which is a command line interface of Ethereum, through shh function of Whisper API.

The nodes participating in the blockchain network for energy trading are prosumers, consumers and a DSO, depending on their roles. Prosumers produce energy and may sell surplus energy to other consumers within the same microgrid. The DSO, an operator of the blockchain network, is responsible for transmission of energy among participants. It also creates/updates smart contracts for energy trading and manages the blockchain network. Algorithm 1 at the end of Section 4 shows Solidity-style pseudo code of smart contract for energy trading.

Lines 3–5 in Algorithm 1 perform initialization. A participating node *X* (*X* = $P_{i}$, $C_{j}$, or DSO) has a pair of addresses, txAddrX and msgAddrX. For example, prosumer $P_{i}$ has $\mathrm{txAddr}P_{i}$ and $\mathrm{msgAddr}P_{i}$. txAddrX is an externally owned account of Ethereum reserved for *X*. Only *X* may access the account using its private key. msgAddrX is an identity to which other nodes may send encrypted messages for node *X* using the Whisper protocol [23]. It allows a node to send messages to other nodes without going through the Ethereum blockchain, saving time-consuming consensus. Transactions to txAddrX should go through the Ethereum blockchain and hence suffer latency needed for consensus.

The DSO broadcasts a smart contract code for energy trading to the blockchain for a microgrid. The smart contract is mined into a block, creating a smart contract account smcAddr (Line 6). Every participating node may send valid transactions to the smcAddr for executing any smart contract functions.

### 4.1. Prevent Tampering of Transaction Records

Each transaction involving an externally owned account should include a digital signature (using a private key) from the owner of the account. The owner is responsible for the transaction it sends and the digital signature guarantees authentication and non-repudiation. The transaction is mined into a block when verified as valid [24]. The transaction in a mined block is regarded as immutable through a mathematical proof [20].

### 4.2. Prevent Double Sale of Energy

Double sale of energy refers to the case in which a malicious node tries to sell the same energy twice or more. To prevent double sale of energy, our system keeps the states of the energy inside a smart contract. The smart contract specifies which nodes are qualified to update which states under what conditions. Thus, updating of the states is only possible through the private key(s) of the qualified node(s). There is no way that hackers may possibly tamper these states without compromising the required private key(s).

For example, the amount of energy injected by prosumer ($P_{i}$) can be updated only by the DSO using its private key after it physically checks the amount of injected energy with its smart meter. Then the $P_{i}$ may send a transaction of “Request_to_Sell” with a specified amount of energy which is equal to or less than the “injected” energy. A private key of prosumer $P_{i}$ is needed for this update transaction to the smart contract. Now that amount of energy changes its state from “injected” to “board”.

Each participant has a state vector representing its own energy as illustrated in Figure 6. The state vector is implemented as an array called energy ownership structure. The state vector is kept inside a smart contract and hence can only be updated by the qualified node(s) or the smart contract itself. In the following subsection, we describe a state diagram which shows the changes in the state vectors of prosumers and consumers.

#### 4.2.1. Implementation of State Diagram

We use enum, a data type defined in Solidity [15], to implement state diagrams in Figure 3 and Figure 4. Enum allows us to define a set of elements to be allowed in the energy state vector. Using this feature, we define five energy states in the state vector for each participant: “register”, “injected”, “board”, “match” and “purchased”. In line 8 of Algorithm 1, an enum data type is declared with the above five states. When a prosumer *i* or a consumer *j* first joins the blockchain network, it should register itself by sending a “Register” transaction to the smart contract (lines 12–19 of Algorithm 1). The “Register” procedure initializes the states of prosumer *i* and consumer *j* to $P_{i}\left(0\right)$ and $C_{j}\left(0\right)$, respectively. The amounts of energy in all the state vectors are initialized to zero. $\left\{I_{i}\left(0\right),\;B_{i}\left(0\right),\;M_{i}\left(0\right)\right\}$ = {0, 0, 0} and $\left\{B_{j}\left(0\right),\;M_{j}\left(0\right),\;P_{j}\left(0\right)\right\}$ = {0, 0, 0}.

When prosumer *i* injects the surplus energy $E_{i}$ through the INJECTENERGY procedure (lines 21–26), its state changes to $P_{i}\left(1\right)$ and $I_{i}\left(1\right)$ is set to $E_{i}$. When prosumer *i* decides to sell the amount $S_{i}\left(t\right)$ in this round of trading, it should send a transaction which executes the “RequestSell” function (line 30). The state of prosumer *i* transitions to $P_{i}\left(2\right)$ and $B_{i}\left(2\right)$ (energy on board for sale) becomes $S_{i}$ and $I_{i}\left(2\right)$ is set to $E_{i}\;-\;S_{i}\left(t\right)$. When the DSO starts a matching phase by sending a transaction to execute the “Matching” function (line 41), the state of prosumer *i* changes to $P_{i}\left(3\right)$.

$M_{i}\left(3\right)$ is set to $S_{m}{}_{i}$, the amount of energy actually matched to be sold, which is less than or equal to $S_{i}\left(t\right)$. $S_{m}{}_{i}$ is computed in lines 42–49. If $S_{i}\left(t\right)\;-\;S_{m}{}_{i}$ is greater than zero, we have$\;I_{i}\left(3\right)=E_{i}\;-\;S_{m}{}_{i}$.

When the matching is complete and the DSO executes the “Trade” function (line 57), prosumer *i* receives payment $S_{m}{}_{i}\cdot p\left(t\right)$ from the smart contract (line 59) and its state changes to $P_{i}\left(4\right)$. Then the prosumer can either inject the remaining energy again or put it on board for sale.

Consumer *j* sends a transaction that executes the “RequestBuy” function (line 35) to state its intention to purchase the amount of $D_{j}\left(t\right)$. If the transaction is confirmed valid, the consumer deposits settlement cost $D_{j}\left(t\right)\cdot p_{max}$ to the smart contract address by sending a transaction that executes the “Transfer” function (line 37). Then the state of consumer *j* changes to $C_{j}\left(1\right)$ and $B_{j}\left(1\right)$ is set to $D_{j}$. When the matching phase is started by the “Matching” function (line 41), the state changes to $C_{j}\left(2\right)$. The amount of energy in the state vector $M_{j}\left(2\right)$ is set to $D_{m}{}_{j}$ as computed in lines 42–49.

When the matching is complete and the DSO executes the “Trade” function (line 57), the consumer receives $D_{j}\left(t\right)\cdot p_{max}$
−
$D_{mj}\cdot p\left(t\right)$ as change from the smart contract (line 62) and the state changes to $C_{j}\left(3\right)$. The consumer receives the energy ownership that it has purchased and the amount of energy is set to $D_{m}{}_{j}$. The consumer can receive the energy from the DSO by using this ownership and its state changes to $C_{j}\left(4\right)$.

#### 4.2.2. Energy Ownership Structure

We choose to represent the states of energy for prosumers and consumers in arrays. Each array represents a prosumer or a consumer. On registration, the first element is created and the “state” of the element is set to “register”. In lines 9–19, the energy ownership structure that includes account, amount, state and timestamp is declared in an array $O_{pi}$ for prosumer *i* (line 13) and $O_{cj}$ for consumer *j* (line 17). Figure 6 illustrates changes in the energy ownership for prosumer 1 and consumer 1 in accordance with trading procedures. The changes of energy ownership are recorded in the smart contract as in Figure 6.

All prosumers and consumers must have an element with the state of “register” as the first element of the array to participate in energy trading. A transaction for executing the smart contract function appends, changes or deletes subsequent elements in accordance with the trading procedures. As shown in Figure 6a, when prosumer 1 injects the energy, an element with the state of “injected” is appended to the array by the DSO (line 25). As shown in Figure 6b, when prosumer 1 makes a request to sell the energy, an element with the state of “board” is appended (line 32). At the same time, the amount of the element with the state of “injected” is reduced accordingly (line 33). As shown in Figure 6c, when the “Matching” function is executed, the element with the state of “board” is deleted (line 50). Elements with the state of “match” are appended to the prosumer’s array (line 51) and consumer’s array (line 53). A new element with the state of “injected” for unmatched energy is appended (line 52). Finally, as shown in Figure 6d, when the “Trade” function is executed, elements with the state of “match” are deleted (lines 60 and 63) and an element with the state of “purchased” is appended (line 64).

In addition, as shown in Figure 7, elements with the state of “injected” or “purchased” exist in multiple elements of the array. Aggregation is performed (line 29) $O_{pi}$ and $O_{cj}$.

#### 4.2.3. Prevent Double Sale Using Smart Contract

We prevent double sale of energy by keeping the energy states inside a smart contract as shown in Figure 8. Smart contract functions are executed only when a calling transaction meets the conditions preset in the smart contract. The conditions may designate sender, parameter and previous states, et cetera. Assume a malicious node sends two transactions to sell the same energy to two different consumers. After the first transaction is executed, states are changed accordingly and are written into an immutable block. There is no way to undo this block. The second transaction is executed only when the node has enough remaining energy after the first sale. If the remaining energy is not enough, the second transaction is rejected by the smart contract, wasting the sender’s Ethereum gas. Gas refers to the fee required to send a transaction on Ethereum blockchain [4].

In addition, a smart contract event is called when an element is appended, changed and deleted. All nodes can detect the event and execute any functions accordingly. Thus, energy trading can be performed automatically and exactly by a smart contract programmed to implement a given trading procedure. Consequently, our proposed system performs a safe and transparent P2P energy trading in a trustless environment by using a smart contract.

### 4.3. Inject Energy Using Whisper

We assume that a distribution network within a microgrid allows two-way electric transmission between prosumers/consumers and the energy storage system (ESS) in the DSO. Smart meters are sealed to be tamper-proof. For simplicity, we assume there is no power loss in electric transmission. As illustrated in Figure 9, when prosumer *i* injects energy into the ESS, the prosumer uses its own messaging address $msgAddrP_{i}$ to send a message to the DSO’s messaging address, msgAddrD via Whisper [23]. The message contains the prosumer’s account ($txAddrP_{i}$) and the amount of injected energy ($E_{i}$). Upon receipt of the message, the DSO checks whether the energy is injected. If energy is injected as promised in the Whisper message, the DSO sends a transaction to execute “inject” function to the smart contract address smcAddr. The smart contract function is only executed when the qualified sender, as dictated in the smart contract (the DSO in this case), sends an “inject” transaction. The smart contract checks whether the sender of the transaction is truly the DSO by using the require function (line 24) which verifies the digital signature from the DSO’s with its known public key. If the transaction is considered valid by the smart contract, a new element with the state of “injected” and the amount of energy ($E_{i}$) is appended to the array for prosumer *i*.

### 4.4. Matching between Prosumers and Consumers

Prosumers and consumers send requests to sell (supply) and to purchase (demand) during aggregation. The aggregate sum of sell (supply) amounts may not be equal to the aggregate sum of purchase (demand) amounts. We calculate a ratio (q) using the total supply (S) and the total demand (D) as in Equations (7) and (8).
(7)$$ES\left(t\right)=\overset{n_{p}}{\underset{i=1}{\sum }}S_{i}\left(t\right),\;ED\left(t\right)=\overset{n_{c}}{\underset{j=1}{\sum }}D_{j}\left(t\right)\;\;\;S_{i}\left(t\right),D_{j}\left(t\right)\geq 0$$
(8)$$q=ED\left(t\right)/ES\left(t\right)$$

$S_{i}\left(t\right)$ is the amount of energy to sell (supply) requested by prosumer *i*. $D_{j}\left(t\right)$ is the amount of energy to purchase (demand) requested by consumer *j*. $n_{p}$ and $n_{c}$ are the number of prosumers and the number of consumers, respectively.

In the matching procedure, $S_{m}{}_{i}$ is the amount of energy actually sold by prosumer *i* and $D_{m}{}_{j}$ is the amount of energy actually purchased by consumer *j* as in Equations (9) and (10). Note that$\;\sum_{i=1}^{n_{p}}S_{mi}$ is equal to $\sum_{i=1}^{n_{c}}D_{mj}$.
(9)$$S_{m}{}_{i}=\left\{\begin{array}{c} q\cdot S_{i}\left(t\right)\;\;if\;ES\left(t\right)\geq ED\left(t\right)\\ S_{i}\left(t\right)\;\;\;\;\;\;\;\;\;\;Otherwise \end{array}\right.$$
(10)$$D_{m}{}_{j}=\left\{\begin{array}{c} D_{j}\left(t\right)\;\;\;\;\;\;\;if\;ES\left(t\right)\geq ED\left(t\right)\\ D_{j}\left(t\right)/q\;\;\;\;\;\;\;\;\;\;\;Otherwise \end{array}\right.$$

### 4.5. Settlement

We choose to make the settlement between sellers (prosumers) and buyers (consumers) go through a smart contract. The role of the smart contract is similar to escrow. The smart contract is a blockchain-based program that encodes the conditions for fulfillment of an agreement between participants. It enables the agreed procedures to be securely executed without any third party.

Figure 10 illustrates our settlement procedure. The aggregation phase begins by the DSO’s transaction to execute the “Roundstart” function (line 28). All prosumers and consumers make requests to sell or purchase energy by sending transactions to execute the “RequestSell” (line 30) or “RequestBuy” (line 35) functions.

Consumer *j* deposits $D_{j}\left(t\right)\cdot p_{max}$, the maximum possible amount of payment (actual price $p\left(t\right)$ ≤ $p_{max}),$ to the smart contract address smcAddr by executing the “Transfer” function (line 37). The payment in our system uses tokens that follow the standard ERC-20(Ethereum Request for Comment 20) [25]. When matching is complete and the DSO executes the “Trade” function (line 57), prosumer *i* receives the payment $S_{m}{}_{i}\cdot p\left(t\right)$ (lines 59–60) and consumer *j* receives the change $D_{j}\left(t\right)\cdot p_{max}-D_{m}{}_{j}\cdot p\left(t\right)$ (lines 63–64) (performed by updating the corresponding balances on the smart contract). Finally, an element with the state of “purchased” is appended to the consumer’s array to reflect energy purchase.

## 5. Experiment Using a Test Ethereum Blockchain

### 5.1. Creation of Virtual Prosumers and Consumers

The experiment is performed using personas which are virtually created prosumers and consumers. We assume each prosumer only sells the energy in the simulation. We define prosumers and consumers using 4 parameters, respectively: MaxSupply, MinSupply, PriceUp and PriceDown for each prosumer and MaxDemand, MinDemand, PriceUp and PriceDown for each consumer. Table 2 defines the parameters.

It is reasonable for us to assume that prosumers increase/decrease supply and consumers decrease/increase demand on increase/decrease in price with presumed upper limit (MaxSupply, MaxDemand) and a lower limit (MinSupply, MinDemand). For simplicity, we assume prosumer *i* requests to sell $i.Supply\left(t\right)$ at the start of the trading round *t*. The $i.Supply\left(t\right)$ depends on $p\left(t-1\right)$, the matching price in round $\left(t-1\right)$. It is determined as in Equation (11). Consumer *j* requests to purchase $j.Demand\left(t\right)$ which is similarly determined in Equation (12).
(11)$$i.Supply\left(t\right)=\left\{p\left(t-1\right)-i.PriceDown\right\}\times \frac{i.MaxSupply-i.MinSupply}{i.PriceUp-i.PriceDown}+i.MinSupply$$
(12)$$j.Demand\left(t\right)=\left\{j.PriceUp-p\left(t-1\right)\right\}\times \frac{j.MaxDemand-j.MinDemand}{j.PriceUp-j.PriceDown}+j.MinDemand$$

Figure 11 illustrates requested energy ($i.Supply\left(t\right)$ or $j.Demand\left(t\right)$) vs.$\;p\left(t-1\right)$ for prosumer *i* (*i*.PriceUp = 175, *i*.PriceDown = 125, *i*.MaxSupply = 2000, *i*.MinSupply = 500) and a consumer *j* (*j*.PriceUp = 175, *j*.PriceDown = 125, *j*.MaxDemand = 2000, *j*.MinDemand = 500) as two personas.

Figure 12 illustrates the impact of “*k*” in Equation (6) on the convergence to equilibrium ($R\left(t\right)=1$) assuming prosumer *i* and consumer *j* respond to price change as in Figure 11. Figure 12 shows $p\left(t\right)$ vs. $R\left(t\right)$ using Equation (6) with $p_{balance}\;$= 150 and $p_{con}$ = 100 for (a) *k* = 3, (b) *k* = 5 and (c) *k* = 7, respectively. We define convergence area as where $p\left(t\right)$ is within $p_{balance}\pm \;$1%.

For the smallest value of *k* (=3) as in Figure 12a, convergence speed is slowest (it takes the largest number of rounds (=12) to arrive within the convergence area). However, the convergence area is closest to equilibrium, $10^{-0.13}\leq R\left(t\right)\leq 10^{0.13}$. For largest value of *k* (=7) as in Figure 12c, convergence speed is fastest (it takes the smallest number of rounds (=8) to arrive within the convergence area). However, the convergence area is farthest from equilibrium, $10^{-0.25}\leq R\left(t\right)\leq 10^{0.25}$.

For the medium value of *k* (=5) as in Figure 12b, the convergence speed (=10 rounds) and the convergence area lies between these two extremes. This illustrates that our dynamic pricing scheme enables the DSO to adjust “*k”* to make a tradeoff between convergence speed and exactness of balancing (narrowness of convergence area) to suit its purpose. 

### 5.2. Setup of an Experimental Ethereum Blockchain

Raspberry Pi devices are chosen to emulate the 11 nodes participating in the test Ethereum blockchain. The nodes implement virtual prosumers, consumers and the DSO. Each node uses the Geth [22] command line interface (CLI) to access the blockchain network, deploy smart contracts, send a transaction and detect events from smart contracts.

All nodes participating in a private Ethereum blockchain should share a genesis block [22]. In our test the Ethereum blockchain has a DSO, 5 prosumers, 5 consumers (each with an account holding 5 Ethereums) and a smart contract. The source code used in our test can be found at: https://github.com/skj1080/energy_trading (accessed on 8 March 2021).

All nodes participating in the Ethereum blockchain need a smart contract address to execute smart contract functions or to be notified of any events from the smart contract.

### 5.3. Execute Functions or Detect Events in a Smart Contract 

Figure 13 shows prosumers, consumers and DSO access to the Ethereum blockchain via Node.js [26]. The Web3 library in Node.js is used to access the Ethereum Virtual Machine. After a smart contract address is generated, each node can execute a smart contract function by sending a transaction and detect any resulting events declared within the function in accordance with the trading procedure. IoT devices for any controls, including transmission of energy, can be linked to certain events from the smart contract through the Node.js. For example, energy transmission after matching can be performed in an autonomous way with appropriate IoT control, removing human involvement.

The execution of the functions in a smart contract may result in events which can be detected by all participating nodes. The events are usually declared in the smart contract. 

Figure 14 shows our testbed representation made of 5 Raspberry Pis which act as 2 prosumers, 2 consumers and a DSO. Figure 15 shows the GUI built with MATLAB for the DSO in the testbed. It shows requests to sell, requests to purchase from two prosumers, two consumers and the DSO. The DSO can choose “*k*” value and it determines the price curve in the center-right. The graph at the bottom shows the change of price as trading periods progress.

### 5.4. Create Persona from Real Prosumers and Consumers

We create personas from real prosumers and consumers at 5 microgrids in Toronto, Canada [13]. Table 3 shows an average supply per hour by 5 prosumers and Table 4 shows an average demand per hour by 5 consumers. We chose the 24th hour to illustrate how our energy trading is actually implemented in each trading period. Total energy supply from the 5 prosumers is 336 kWh, and total energy demand is 228 kWh. We assume total energy supply is the aggregate from the 5 prosumers in Table 5. Similarly, total energy demand is the aggregate from 5 consumers.

The experiment is performed according to the trading procedure in Section 4. Transaction results are recorded in the form of BigNumber {s: sign, e: exponent, c: value} through event detection. BigNumber is a Javascript library for arbitrary-precision arithmetic [27]. The state of energy is displayed as 0: register, 1: injected, 2: board, 3: match, and 4: purchased. More details about the experiment as follows.

#### 5.4.1. Inject Energy

In this procedure, a prosumer injects the energy into the ESS of the DSO and sends a message to the DSO using Whisper. The DSO receives the message which contains the prosumer’s account and the amount of injected energy. Upon receipt of the message, the DSO sends a transaction to execute the “inject” function after physical checking of injected energy. After that, a new element is appended to the prosumer’s ownership array.

#### 5.4.2. Aggregation

Prosumers and consumers make requests to sell or purchase energy by sending transactions. Five prosumers and five consumers present supply (sell) and demand (purchase) as shown in Table 5.

When a consumer’s transaction is confirmed as valid, each consumer deposits tokens (each consumer’s demand multiplied by the maximum price 130) to the smart contract address.

#### 5.4.3. Matching

After the aggregation, the DSO computes total supply (request to sell), total demand (request to purchase) and the ratio q (total demand/total supply). If total supply is 336 kWh and total demand is 228 kWh as in our example, the ratio q is 0.68. Since total supply (S) is greater than total demand (D) each prosumer is able to sell *q* × S (matched supply) and remainder (unmatched supply) is injected back to itself. Refer to the state diagram in Figure 3. Table 6 shows a result of matching. All matched supply and demand are appended to the ownership array as an element with the state of “match”. All unmatched supplies are appended as an element with the state of “injected” to each array.

#### 5.4.4. Settlement

One single price is used for a trading period. The price is determined by Equation (6) with R(t) is equal to *q* = $ED\left(t\right)/ES\left(t\right)$. If we assume *k* = 3, $p\left(t\right)$ = 98.9 Tokens/kWh. When the DSO sends a transaction to execute the “Trade” function, each prosumer is paid $p\left(t\right)$ times the matched supply. Each consumer is refunded its deposit minus matched demand times $p\left(t\right)$. The results are shown in Table 7.

### 5.5. Ethereum Gas Cost

Table 8 shows Ethereum gas (gas) consumption for major functions in our implementation along with their monetary cost in US dollars (if we assume our system runs on the public Ethereum network). As of 4 January 2021, 1 ether costs around $969 and the average price of unit gas is 190 Gwei (190 × 10^−9^ ether) according to Etherscan [28]. This estimate of gas consumption serves as a measure for the complexity of our system. The Deployment function is performed only once at the start of the trading system while other functions may be performed in each round.

In Table 9, we show total gas consumption during a period in our implementation. We also estimate total gas consumption for $n_{p}$ prosumers and $n_{c}$ consumers. However, the gas consumption per user remains the same.

Table 10 shows that the total gas consumption in our system is smaller than Galal and Youssef [29] by 41%. Note that our system removes the ZKP(Zero Knowledge Proof) related functions needed for security of bidding in Galal and Youssef [29]. 

Figure 16 compares a progressive gas consumption of our system against Galal and Youssef [29] with 5 prosumers and 5 consumers for up to 10 rounds. As the number of trading rounds increase, saving of gas in our system against Galal and Youssef [29] increases by as much as 78%.

Table 11 shows how to estimate needed storage as a function of participants ($n_{p}$ and $n_{c}$). In Ethereum, there are $2^{256}$ different keys and each key can store 32 bytes. So, that is a total of $2^{261}$ bytes that could be stored [15]. We may store as many as $2^{261}$ bytes inside an EVM. We need ($n_{p}$ × 468 + $n_{c}$ × 340 + 2144) bytes for trading with $n_{p}$ prosumers and $n_{c}$ consumers. The constraint for the number of prosumers and in terms of storage can be represented as in Equation (13).
(13)$$2^{261}\geq n_{p}\;\times \;468\;+\;n_{c}\;\times \;340\;+2144$$

The storage capacity required in our system increases with the number of participants. The following table shows the smart contract storage used for each participant.

The storage requirement increases by 468 bytes per prosumer and 340 bytes per consumer. The storage limit of the smart contract is $2^{261}$. In this case, there is a big difference between storage requirement and storage limitation. Therefore, we expect that there will be no problem in terms of storage in our system.

In R. Skowronski [16], the number of transactions for Bitcoin is derived as in Equation (14)
(14)$$\tau =(\frac{\beta }{\mu }\;\times \;\frac{\rho }{\theta })$$
where β,  μ, ρ and θ denote block size, transaction size, block creation interval and time frame. The transactions per second (tps) can be obtained by dividing τ by θ. Our trading system is built on Ethereum and the number of transactions is usually limited by the maximum allowable gas per block [4].

Assume we have $n_{p}$ prosumers and $n_{c}$ consumers. Each prosumer consumes $g_{p}$ gas and each consumer consumes $g_{c}$ gas per trading period. The gas limit per block is $g_{b}$. Then we need $\left(g_{p}n_{p}+g_{c}n_{c}\right)/g_{b}$ blocks to perform one trading period. If we assume ρ to be a block interval time in seconds, we may represent trading matches per second as in Equation (15)
(15)$$\frac{g_{b}}{\left(g_{p}n_{p}+g_{c}n_{c}\right)}\;\times \;\frac{1}{\rho }$$
where $n_{p}$ and $n_{c}$ denote the number of prosumers and consumers, respectively. We multiply 86,400 (24 × 60 × 60) to the number in Equation (15) to obtain the maximum number of trading matches per day. Recently, the maximum gas limit per block, $g_{b}$, is usually set to 12,500,000 [28]. In our trading system experiment, $g_{p}$ is around 200,000 and $g_{c}$ is around 30,000.

In our current implementation, we have only a few states for which we have enumerated all possible state changes. So, we do not expect any inevitable disturbances. However, if unthinkable technical faults erupt, then the gas limit is the last resort. We limit gas allowance to only covering valid transactions to avoid fatal damage. To deal with unexpected changes in demand, we only allow manageable numbers of prosumers and consumers to join our trading system on a private Ethereum blockchain.

## 6. Conclusions

In this paper, we propose a smart contract-based P2P energy trading system with a dynamic pricing model. The smart contract resides on the blockchain shared by participants and hence guarantees exact execution of trade and keeps immutable transaction records. It removes high costs and overheads needed against hacking or tampering in traditional server-based P2P energy trade systems.

Double sale is prevented by maintaining the state of energy inside the smart contract. Furthermore, we create a dynamic pricing model which enables the DSO to make a trade-off between convergence speed and exactness of balancing between supply and demand within a microgrid.

We create personas participating in energy trading and conduct virtual simulations on a testbed with 5 prosumers, 5 consumers and the DSO. Our system saves gas needed to operate by as much as 78% compared with a known bidding style energy trading system on blockchain.

## Figures and Tables

**Figure 1 sensors-21-01985-f001:**
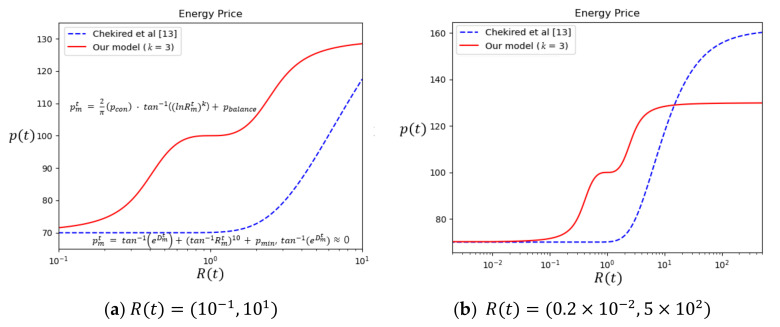
Comparison of our pricing model (solid red line) with (*k* = 3) against Chekired et al. [13] (dotted blue line), with $p_{balance}\;$= 100, $p_{con}$ = 30. (**a**) $R\left(t\right)=\left(10^{-1},10^{1}\right)$ (**b**) $R\left(t\right)=\left(0.2\times 10^{-2},5\times 10^{2}\right)$.

**Figure 2 sensors-21-01985-f002:**
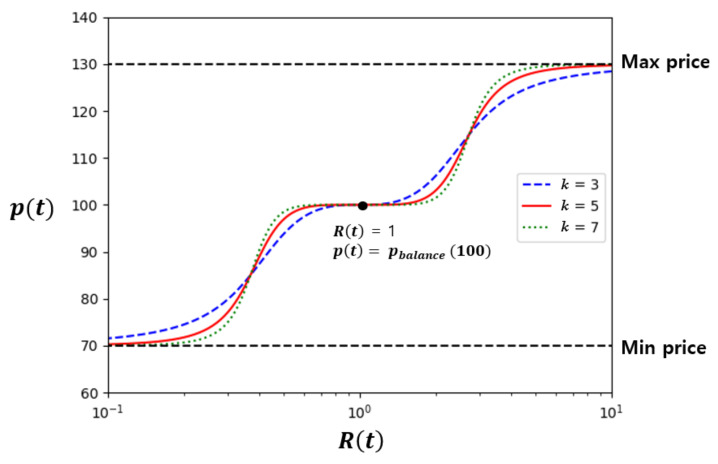
The proposed dynamic pricing with varying *k* ($p_{balance}=100,\;p_{con}=30)$.

**Figure 3 sensors-21-01985-f003:**
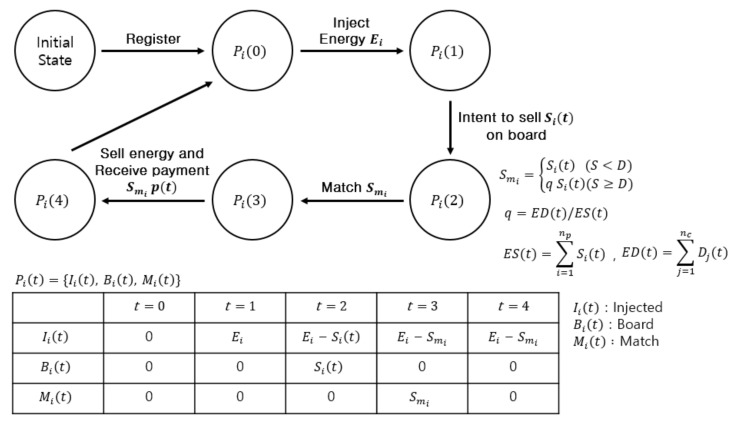
A state diagram for prosumer *i* in phase *t* = 0, 1, 2, 3 and 4 of energy trading.

**Figure 4 sensors-21-01985-f004:**
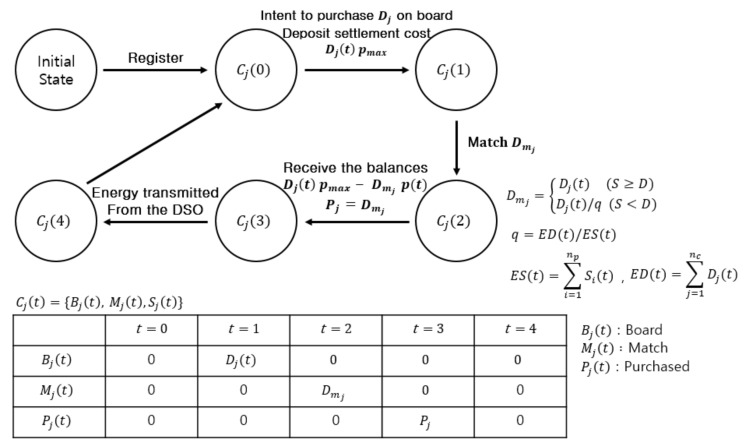
A state diagram for consumer *j* in phase *t* = 0, 1, 2, 3 and 4 of energy trading.

**Figure 5 sensors-21-01985-f005:**
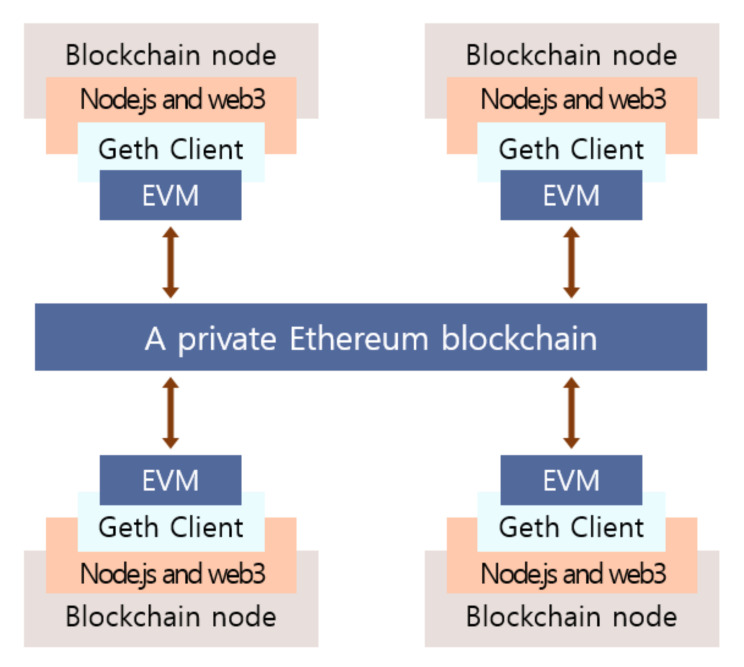
The software architecture of proposed peer-to-peer (P2P) energy trading implementation on a private Ethereum blockchain.

**Figure 6 sensors-21-01985-f006:**
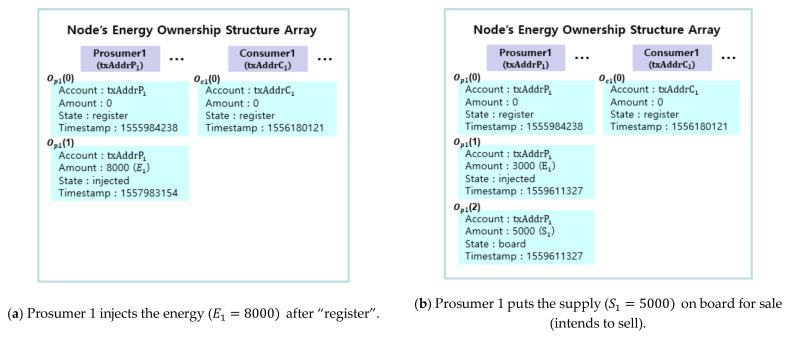
Illustration of change in the energy ownership in accordance with trading procedures.

**Figure 7 sensors-21-01985-f007:**
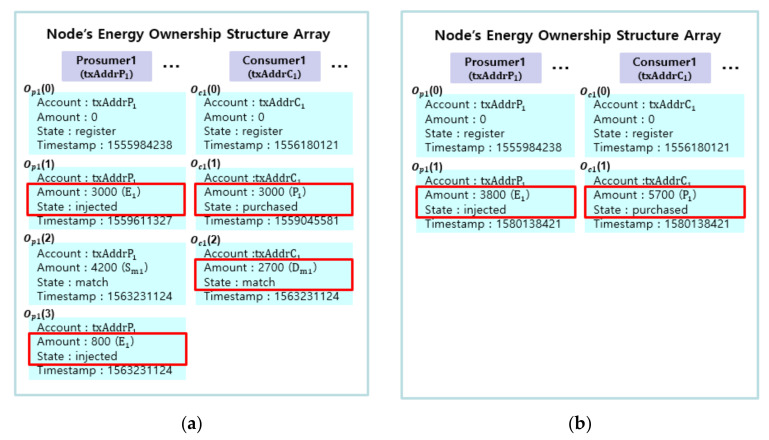
Example of aggregation of multiple elements: (**a**) Before aggregation and (**b**) after aggregation.

**Figure 8 sensors-21-01985-f008:**
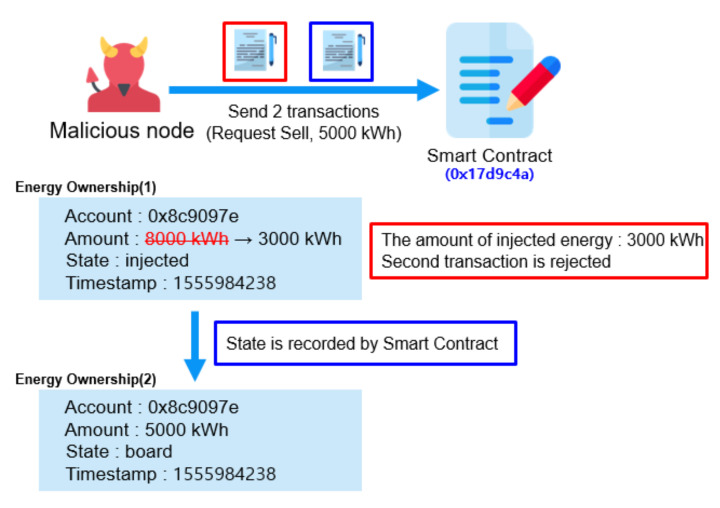
Prevent double sale by keeping the states of energy inside a smart contract.

**Figure 9 sensors-21-01985-f009:**
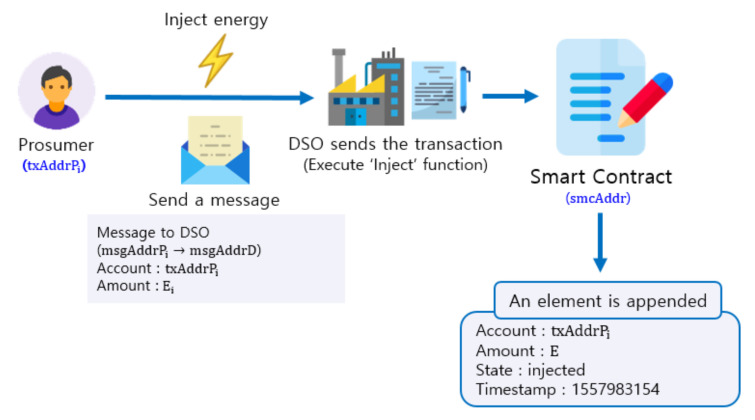
Appending an element to the energy ownership structure after energy injection.

**Figure 10 sensors-21-01985-f010:**
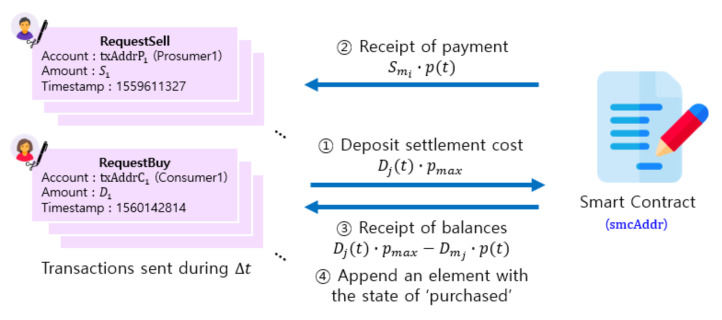
The process of payment between prosumers and consumers in the energy trading.

**Figure 11 sensors-21-01985-f011:**
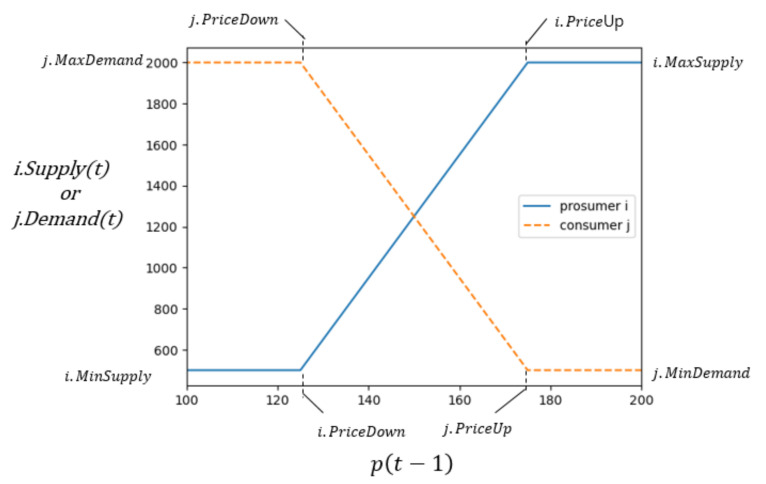
An illustration of ($i.Supply\left(t\right)$ or $j.Demand\left(t\right)$) vs. $p\left(t-1\right)$.

**Figure 12 sensors-21-01985-f012:**
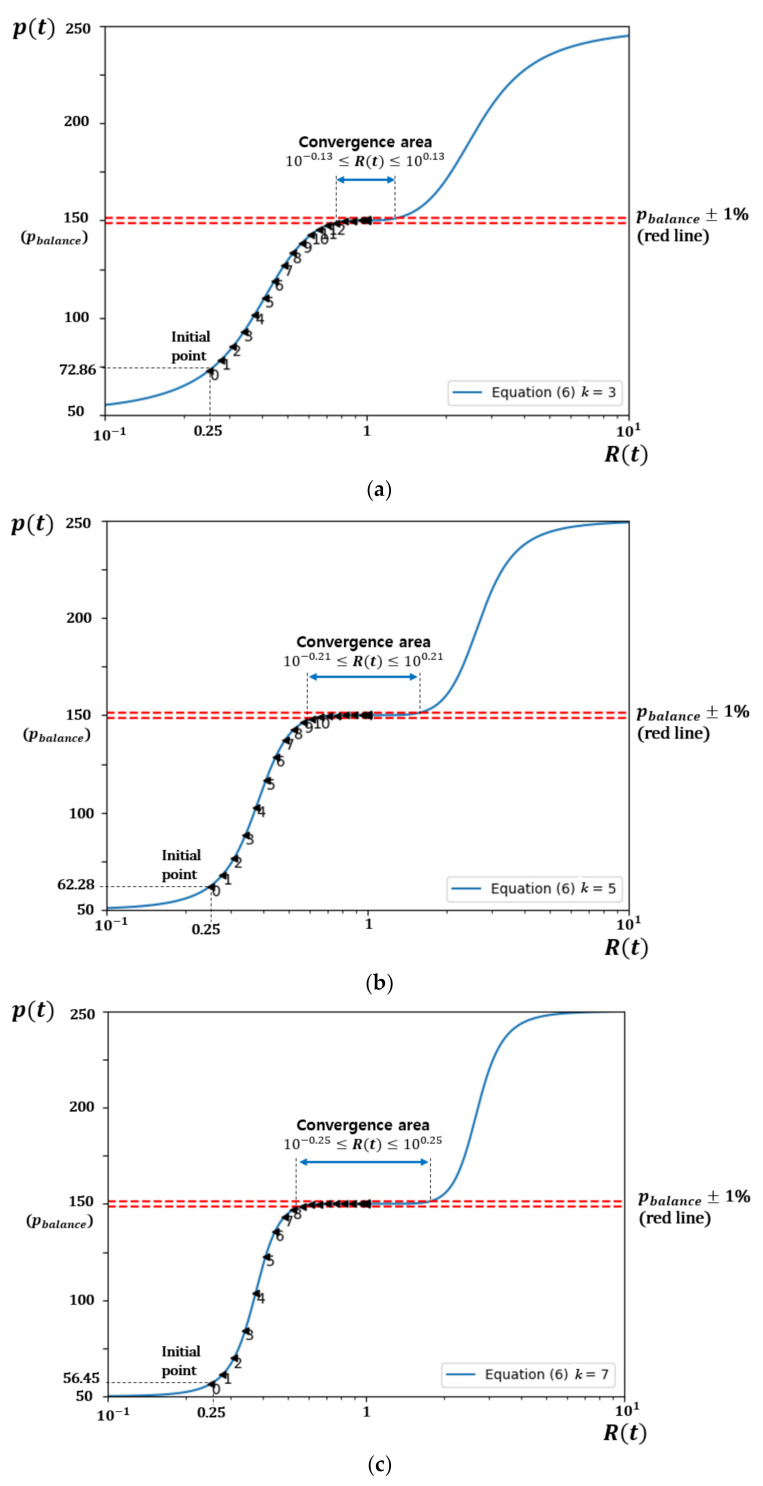
$p\left(t\right)$ vs. $R\left(t\right)$ using Equation (6), $p_{balance}\;$= 150 and $p_{con}$ = 100, (**a**) *k* = 3, (**b**) *k* = 5 and (**c**) *k* = 7.

**Figure 13 sensors-21-01985-f013:**
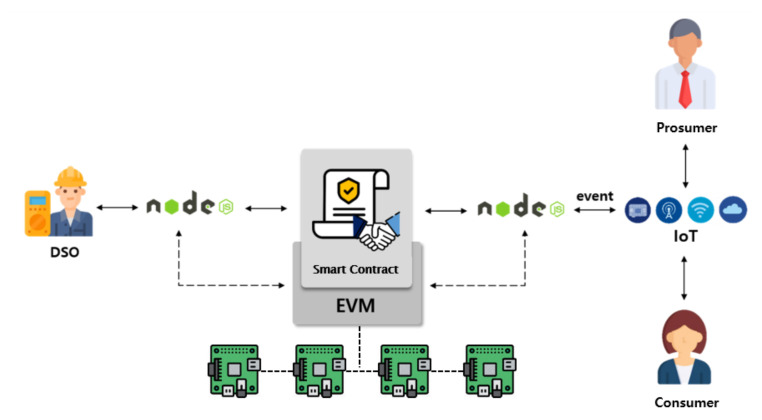
Our testbed for experiment of our P2P energy trading on Ethereum blockchain.

**Figure 14 sensors-21-01985-f014:**
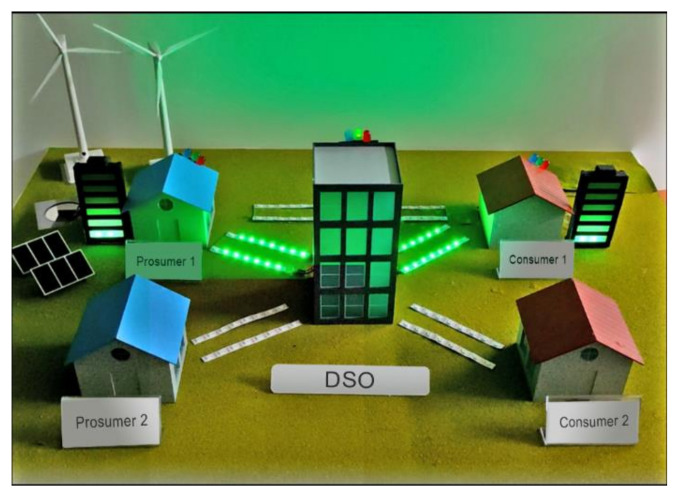
Testbed for experimentation.

**Figure 15 sensors-21-01985-f015:**
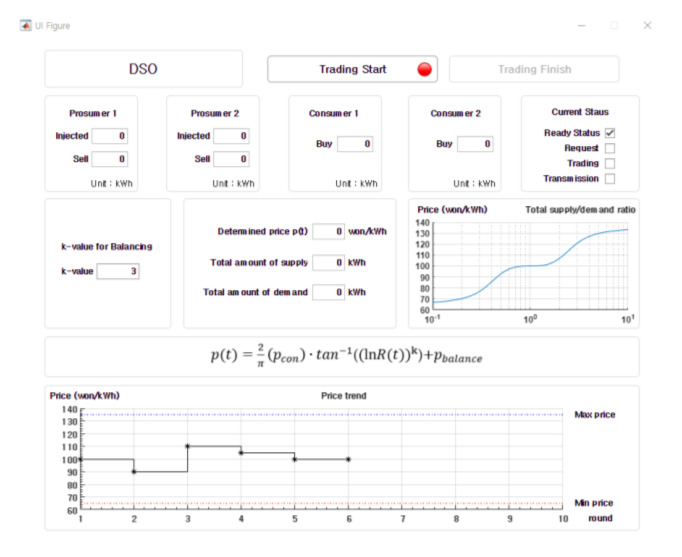
A GUI built with MATLAB for experimentation.

**Figure 16 sensors-21-01985-f016:**
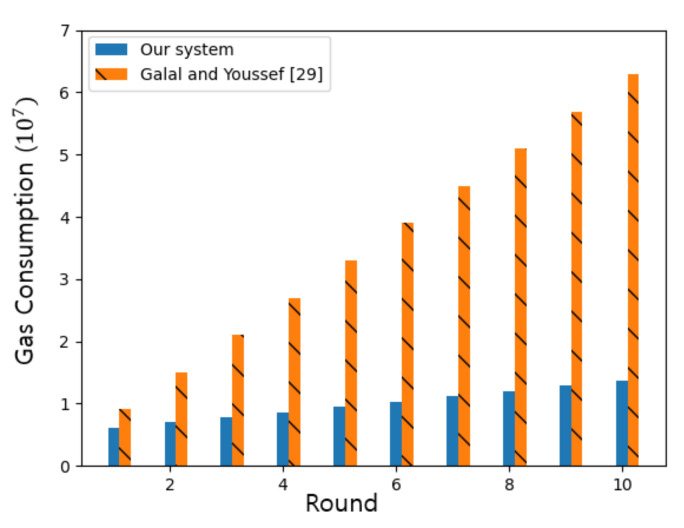
Progressive gas consumption of our system compared against Galal and Youssef [29].

**Table 1 sensors-21-01985-t001:** Notations for energy trading algorithm.

txAddrX	An externally owned account in Ethereum for X
msgAddrX	An address for X in Ethereum messaging Whisper
txAddr $P_{i}$	Transaction address of the *i*-th producer
msgAddr $P_{i}$	Message address of the *i*-th producer
txAddr $C_{j}$	Transaction address of the *j*-th consumer
msgAddr $C_{j}$	Message address of the *j*-th consumer
$E_{i}$	Energy amount injected by prosumer *i*
$O_{pi}$	Producer’s energy ownership structure
$O_{cj}$	Consumer’s energy ownership structure
$S_{i}$	Amount of intent to sell
$S_{mi}$	Matched amount of $S_{i}$
$D_{j}$	Amount of demand to buy
$D_{mj}$	Matched amount of $D_{j}$

**Table 2 sensors-21-01985-t002:** Parameters for personas representing prosumers and consumers.

**Parameter (Postfix)**	**Description**
*i*.MaxSupply	The maximum supply that prosumer *i* can request to sell
*i*.MinSupply	The minimum supply that prosumer *i* can request to sell
*j*.MaxDemand	The maximum demand that consumer *j* can request to buy
*j*.MinDemand	The minimum demand that consumer *j* can request to buy
*i*.PriceUp, *j*.PriceUp	Highest price (No more increase/decrease in supply/demand after the price reaches this value)
*i*.PriceDown, *j*.PriceDown	Lowest price (No more increase/decrease in supply/demand after the price reaches this value)

**Table 3 sensors-21-01985-t003:** Average total energy supply per hour per day from 5 chosen microgrids.

Average Energy Supply per Hour												
Hour	1	2	3	4	5	6	7	8	9	10	11	12
kWh	388	386	401	416	442	468	503	538	573	608	637	665
Hour	13	14	15	16	17	18	19	20	21	22	23	24
kWh	696	727	736	745	738	731	695	658	596	534	435	336

**Table 4 sensors-21-01985-t004:** Average total energy demand per hour per day from 5 chosen microgrids.

Average Energy Demand per Hour												
Hour	1	2	3	4	5	6	7	8	9	10	11	12
kWh	336	324	330	335	353	370	406	442	493	544	580	616
Hour	13	14	15	16	17	18	19	20	21	22	23	24
kWh	651	686	702	718	707	696	657	618	530	442	335	228

**Table 5 sensors-21-01985-t005:** Supply and demand presented by prosumers and consumers.

Prosumer	Supply	Consumer	Demand
1	71 kWh	1	50 kWh
2	55 kWh	2	53 kWh
3	60 kWh	3	35 kWh
4	100 kWh	4	60 kWh
5	50 kWh	5	30 kWh
Total	336 kWh	Total	228 kWh

**Table 6 sensors-21-01985-t006:** Matching results for all prosumers and consumers.

Prosumer	Matched Supply	Unmatched Supply	Consumer	Matched Demand
1	48 kWh	23 kWh	1	50 kWh
2	37 kWh	18 kWh	2	53 kWh
3	41 kWh	19 kWh	3	35 kWh
4	68 kWh	32 kWh	4	60 kWh
5	34 kWh	16 kWh	5	30 kWh
Total	228 kWh	108 kWh	Total	228 kWh

**Table 7 sensors-21-01985-t007:** The number of tokens paid to prosumers and the refund to consumers.

Prosumer	Paid	Consumer	Refund
1	4747.2	1	1555
2	3659.3	2	1648.3
3	4054.9	3	1088.5
4	6725.2	4	1866
5	3362.6	5	933
Total	22,549.2	Total	7090.8

**Table 8 sensors-21-01985-t008:** Gas consumption for the functions in our system assuming the public Ethereum network.

Function	Gas Consumption	Gas Cost ($)
Deployment	3,003,112	552.90
Register	160,284	29.51
Inject	131,679	24.24
Request to Buy	23,523	4.33
Request to Sell	23,591	4.34
Matching	354,520	65.27
Transfer	25,574	4.71

**Table 9 sensors-21-01985-t009:** Total gas consumptions per trading period.

Total Gas Consumptions	
4 Prosumers and 4 Consumers	$n_{p}$ $\;\mathrm{Prosumers},\;n_{c}\;\mathrm{Consumers}$
3,003,112	3,003,112
1,282,272	($n_{p}$ + $n_{c}$) × 160,284
526,716	$n_{p}$ × 131,679
94,092	$n_{c}$ × 23,523
94,364	$n_{p}$ × 23,591
354,520	354,520
204,592	($n_{p}+n_{c}$) × 25,574
5,559,668	3,357,632 + $n_{p}$ × 341,128 + $n_{c}$ × 209,381

**Table 10 sensors-21-01985-t010:** Comparison of our system against Galal and Youssef [29] in terms of gas consumption.

Our System		Galal and Youssef [29]	
Function	Gas Consumption	Function	Gas Consumption
Deployment	3,003,112	Deployment	3,131,261
Register	160,284	Bid	130,084
Inject	131,679	Reveal	132,849
Request to Buy	23,523	ClaimWinner	166,288
Request to Sell	23,591	ZKPCommit	656,689
Matching	354,520	ZKPVerify	2,002,490
Transfer	25,574	VerifyAll	46,580
		Withdraw	47,112
Total	3,722,283	Total	6,313,353

**Table 11 sensors-21-01985-t011:** Need storage as a function of participants.

Participant	Contents	Per Participant	$n_{p}\;\mathrm{Prosumers}\;\mathrm{and}\;$ $n_{c}\;\mathrm{Consumers}\;\mathrm{Size}$
Prosumer	Energy ownership states (5 key/value pairs) + Account	5 × 64 bytes + 52 bytes = 372 bytes	$n_{p}$ × 372 bytes
Consumer	Energy ownership states (3 key/value pairs) + Account	3 × 64 bytes + 52 bytes = 244 bytes	$n_{c}$ × 244 bytes
DSO	Management (ex. Price Setting)	2 KB	2 KB
Smart Contract	Registration of participants	96 bytes per participant	($n_{p}$ + $n_{c}$ + 1) × 96 bytes
Total	-	-	($n_{p}$ × 468 + $n_{c}$ × 340 + 2144) bytes

## Data Availability

Not applicable.

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
