# Peer review of "A Smart Contract-Based P2P Energy Trading System with Dynamic Pricing on Ethereum Blockchain"

_sensors, 2021, doi:10.3390/s21061985_

Round 1

Reviewer 1 Report

The paper presents a blockchain based peer to peer market for the balancing of a microgrid. Using a special market algorithm, and based on the Ethereum blockchain currency, it manages the exchange of contracts for the microgrid and the payment side. 

This is an interesting topic, and it is well presented, including an experiment model study. It is nice to see an actual implementation, although it would be nice to see a bit more detail, such as LOC, computational complexity etc, or maybe even key parts of the source code. 

In terms of improvements, I would certainly introduce some of the terms more carefully. "Prosumer" has several definitions, and the one here is producer/consumer. Equally, "gas" can refer to different thing, especially when talking about energy. Here, it refers to the transaction cost of Ethereum transactions. 

Finally, there are a few questions that remain unanswered about potential applications. The system does seem to require the participation of a DSO as the moderator of the market - so how is this different from a DSO running a traditional energy market? And how would this system ensure that the promised transactions are actually delivered in the real world, that the right amount of power is provided or withdrawn? Finally, how would this system deal with inevitable disturbances causes by technical faults or unexpected changes in demand?

That being said, these are minor issues, and the paper is generally sound. 

Reviewer 2 Report

One of the downsides I have found while reviewing is the issue of privacy and second, of the overhead imposed on the decentralized state machine as authors have chosen to rely on the smart contract as their main computational and data-storage apparatus.

As for privacy there’s no way to introduce encryption on data flowing to a smart contract (assuming it needs to know the decryption key)thus the requests by parties are open and meta-data regarding these may be collected. In my opinion this deserves to be noted as the very subject of privacy in regard to energy trade is an active field of research.

The article is very interesting, especially thanks to a concrete implementation. I would suggets authors dropping in few lines on the forseebale performance (matched transactions per day) as a function of the number of trade participants, as well as, the storage overhead within the VM as the function of the number of users along with associates Gas costs.

I can see authors depicted prices for particular operations however, in my opinion that is not enough for depicting feasibility of the entire conceive on a larger scale to the reader.

I am a curious to see how the current implementation of the Ethereum state-machine (the possible number of state-transitions per second) and the current economy around its chain would cope with that.

Authors seem to rely heavily on the decentralized state-machine for computation ( transaction matching) and storage of data-related to power-flows.

Authors could attempt facilitating the very payments between peers off-the-chain for pre-longed periods of time during power flows and cashing out these on the chain when finalized, yet again to lower the burden on the decentralized state-machine (just an idea).

When we consider the VM as a true infinite state-machine, nothing of what I’ve mentioned would be needed. Hower, in reality, these problems of achievable throughout and associated costs need to be addressed / described matched especially seeing authors have already chosen a particular VM as authors do well understand by depicting at least the costs of unitary operations.

After reading the article further it turns out that additionally smart contract is used additionally for storing the history of power-flows for double-spend detections. This yet again asks for analysis of performance and associated costs.

In related works authors could consider mentioning “On the applicability of the GRIDNET protocol to Smart Grid environments” which as far as I’m concerned proposed open-trade through blockchain and a hierarchy-based control of flows for the first time probably also “the open blockchain-aided multi-agent symbiotic cyber–physical systems” by same author which tackles the problematics of aiding cyber-physical systems through blockchain-based VMs.

I would advise for acceptance of the article after the above suggestions have been addressed. The overall structure of the article is very good, the implementations is very interesting. Still, as authors have already tackled practice and since we all are interested in practicality of the conceives a more detailed picture of the overall feasibility and costs on a larger scale is needed.

Reviewer 3 Report

The paper presents a peer-to-peer energy trading system based on dynamic pricing and blockchain technology to automate trading operations and prevent double sales. The authors describe the proposed dynamic pricing system, the blockchain definition, and a tentative validation with experiments on a testbed simulation based on real data.

The paper is well structured, especially the parts describing the related works, the proposed dynamic pricing modeling, and the blockchain definition. The only drawback I found in the paper is the description of the experiment implementation, which is not common in this kind of papers. The authors should give more emphasis to the experimental setup and the discussion of the results, instead of writing too many implementation details in Section 5, with a lot of screenshots of the terminal and the source code which do not add anything to the scientific soundness of the paper.

I think that instead of showing screenshots with code and terminal executions, the authors might evaluate to: 
a) publish a video (to be linked in the paper) where they show and comment the execution of the experiments, recording their terminal. In this way, instead of describing the programming of the experiment, they would leave only the results and the relative discussion in the paper and the readers interested in the implementation would follow the video link;
b) share the experiment source code in a public repository, and put the link in the paper.

None of these two points is mandatory. I leave to the authors the choice between publishing:

a) the video;
b) the source code;
c) both;
d) none of the two. 

However, I suggest them to be more concise in describing the implementation in the paper.

---
Minor comments:

1. The related works section is quite deep and gives the necessary background of the presented research. However, being very long, I suggest to separate it in subsections, given that, even in the current version, the authors clearly identify three topics: a) pricing models b) blockchain technology c) use of blockchains in energy trading. Therefore, my suggestion is to structure the related works section into three subsections following these three topics, in order to help the reader in better understanding the related research.

2. Row 365: solidity complier -> solidity compiler

3. there is no point of having some figures with the command line or the code: they do not add anything to what the authors explain. For example:
a) there is no need to use a figure (13) to claim that in the simulation there are 1 DSO, 5 prosumers and 5 consumers;
b) the implementation details of the experiment programming do not add anything to the significance of the experiments (Figures 15, 16, and 17 just list some variables and method calls and can be removed);
c) Figures 20 and 21 do not add anything to the explanation of the energy injection;
d) Figures 22, 23, 24, 25, 26, 27 do not add much to the experiment explanation. To a limited extent, those images just show that the experiment was actually executed, but they are not necessary.

4. I guess this will be fixed by MDPI staff in case of acceptance, but there are many formatting problems in the paper: most of the figures, tables and the algorithm are beyond the margins or not aligned with the text margins.

5. The authors should use the terms of the CRediT taxonomy in the "Author Contributions" section. See https://img.mdpi.org/data/contributor-role-instruction.pdf
